# Two-layer Electrospun System Enabling Wound Exudate Management and Visual Infection Response

**DOI:** 10.3390/s19050991

**Published:** 2019-02-26

**Authors:** Mohamed Basel Bazbouz, Giuseppe Tronci

**Affiliations:** 1Textile Technology Research Group, School of Design, University of Leeds, Leeds LS2 9JT, UK; 2Biomaterials and Tissue Engineering Research Group, School of Dentistry, St. James’s University Hospital, University of Leeds, Leeds LS9 7TF, UK

**Keywords:** infection, colour change, fibres, free surface electrospinning, bromothymol blue, core-shell fibres

## Abstract

The spread of antimicrobial resistance calls for chronic wound management devices that can engage with the wound exudate and signal infection by prompt visual effects. Here, the manufacture of a two-layer fibrous device with independently-controlled exudate management capability and visual infection responsivity was investigated by sequential free surface electrospinning of poly(methyl methacrylate-co-methacrylic acid) (PMMA-co-MAA) and poly(acrylic acid) (PAA). By selecting wound pH as infection indicator, PMMA-co-MAA fibres were encapsulated with halochromic bromothymol blue (BTB) to trigger colour changes at infection-induced alkaline pH. Likewise, the exudate management capability was integrated via the synthesis of a thermally-crosslinked network in electrospun PAA layer. PMMA-co-MAA fibres revealed high BTB loading efficiency (>80 wt.%) and demonstrated prompt colour change and selective dye release at infected-like media (pH > 7). The synthesis of the thermally-crosslinked PAA network successfully enabled high water uptake (*WU* = 1291 ± 48 − 2369 ± 34 wt.%) and swelling index (*SI* = 272 ± 4 − 285 ± 3 a.%), in contrast to electrospun PAA controls. This dual device functionality was lost when the same building blocks were configured in a single-layer mesh of core-shell fibres, whereby significant BTB release (~70 wt.%) was measured even at acidic pH. This study therefore demonstrates how the fibrous configuration can be conveniently manipulated to trigger structure-induced functionalities critical to chronic wound management and monitoring.

## 1. Introduction

In chronic wounds, such as leg and diabetic foot ulcers, infection requires antibiotic therapy and frequent hospitalisation, and may, in some cases, lead to surgery [1,2,3,4]. With the spread of antimicrobial resistance, infection causes increased wound chronicity and risks of gangrene and amputation, creating a burden on patients and healthcare providers worldwide [5,6,7]. Continuous chronic wound monitoring is therefore a promising strategy to enable early detection of infection, aid diagnosis and inform therapeutic decisions [8,9,10]. At the same time, reliable wound monitoring devices are still hardly realised, due to the complex infection mechanisms and healing process of these wounds [9]. Aiming at next-generation wound care, multifunctional cost-effective wound management products are therefore needed to support healing in non-self-healing wounds and to signal the occurrence of infection by e.g., visual effects [8].

Although multiple wound biomarkers have been reported for wound monitoring, such as pH [11], temperature [12,13], neutrophil extracellular traps (NETs) [14,15], hydrogen peroxide [16,17], lactate [18,19], and wound exudate volume [20], pH monitoring has been reported as one of most versatile strategies [7,11]. The exudate of healthy wound is typically acidic (pH = 4–6), whilst it falls to more alkaline values (pH > 7) when wound infection occurs [10]. A pH range of 7.15–8.9 has been measured in infected wounds [21,22], whereby a direct relationship between wound alkalinity and chronicity was observed [23]. On the other hand, as the wound proceeds towards healing, an acidic wound pH has been found to be optimal for tissue granulation to occur [22,24,25]. Numerous efforts have been directed toward the development of pH monitoring systems [26,27], e.g., via ion sensitive field effect transistor (ISFET) [28], fibre optics [29], near infrared spectroscopy [30], nuclear magnetic resonance [31], fluorescent pH indicators and potentiometric pH sensors [32,33]. Despite these efforts, very few systems became approved for medical use due to reliability issues and time-consuming design and incompatibility with wet environment (Table 1) [7,26,34,35,36,37,38,39,40,41,42,43,44,45,46,47,48]. Therefore, the need for an easy-to-apply, fibre-based, and inexpensive system capable to support wound exudate management and display visual infection responsivity would greatly reduce healthcare costs and limit misuse of antibiotics.

Halochromic dyes consist of a pH-sensitive chromophore and respond to environmental pH by a colour change [49]. Depending on the molecular structure of the chromophore, specific colouration effects can be observed at defined ranges of pH. For instance, bromothymol blue (BTB) exhibits a yellow colour when dissolved at a pH lower than 6.0, whilst it triggers either green or blue coloration in alkaline environment [50]. Incorporation of a halochromic dye into a desired fibrous matrix could therefore be exploited aiming to integrate visual infection responsivity and wound exudate management capability. Indeed, the fibrous matrix represents an ideal carrier for halochromic dyes aiming to achieve long-lasting infection responsivity and inherent liquid (i.e., wound exudate) absorption capability relevant to wound healing [51,52,53].

Electrospun (ES) nanofibrous membranes have attracted great attention as effective wound management devices due to their small sized fibres with fine interconnected pores and tunable porosity, extremely large surface area to volume ratio, high absorbance capacity, and the manufacturing compliance with polymers commonly used in wound care products [54,55]. The presence of fibres facilitate liquid transport in the fabric due to large and easily accessible pores among fibres, leading to fast response to biochemical stimuli, making them promising for both advanced wound care and sensing applications [56,57,58]. Moreover, by using concentric spinnerets, fibrous meshes of core-shell fibres can be achieved [59], aiming to control the release kinetics of fibre-encapsulating bioactive agents via variation of the spinneret dimensions.

In light of the straightforward manufacture, encapsulation of halochromic dyes in electrospun fibres was carried out for detecting pH values [51,52,53], and paved the way to a simple and eco-friendly tool for live monitoring of wound infection [58]. On the other hand, while electrospinning is an effective technique for generating nanofibrous membranes, the manufacturing yield is typically restricted to 0.1–1.0 g·h^−1^ for a single spinneret [60,61], so that industrial scale fibre manufacture demands can hardly be met [55,62]. To overcome this limitation and increase the yield of fibre formation, a great deal of attention has been put towards free surface electrospinning (FSES) [63,64,65], which generates high fibre yields via the rotation of a roller surface in a polymer solution. With nonwoven membranes obtained with 50–500 nm fibre diameter at a production rate of 1.5 g·min^−1^ per meter of roller length [66], this mechanism enables high scalability, low cost, as well as easy operation in comparison with nonwoven membranes electrospun from single spinneret [67,68]. Furthermore, FSES allows the sequential deposition of fibrous layers and thus the preparation of multi-layer fibrous configurations, which is key aiming to achieve structure-induced functions to suit specific end use [4,69,70].

In the present study, we investigated the scalable manufacture of a novel “early warning” fibre-based passive system, intended to create visual effects to patients and carers in case of wound infection. The membrane was designed to be flexible and easily assembled into a wearable patch to be customised into a wound dressing, so that visual infection responsivity could be integrated with wound exudate management capability, to enable wound monitoring via simple colour coding. We hypothesised that FSES could offer a scalable manufacturing platform to generate a two-layer fibrous configuration with independently-controlled exudate management capability (provided by the wound contact layer) and visual infection responsivity (introduced by the encapsulation of BTB as the halochromic dye in alkali-soluble fibres). Polyacrylic acid (PAA) was selected as building block of the wound contact layer, since it has already been employed for moisturisation of wound surfaces [71]. Poly(methyl methacrylate-co-methacrylic acid) (PMMA-co-MAA) was on the other hand electrospun to achieve BTB-encapsulated fibres, to enable colour change capability at infection-relevant pH, in light of its well-established applicability in pharmaceutics and selective solubility in alkaline environments [72,73]. The use of PAA, PMMA-co-MAA and BTB was also supported by the fact that these materials are considered non-hazardous substances according to the European regulation EC 1272/2008, so that the infection-induced BTB release would not negatively impact on wound healing. The resulting system was realised via sequential free surface electrospinning of PAA and BTB-loaded PMMA-co-MAA solutions, followed by thermal treatment, aiming to achieve a covalently-crosslinked network in PAA fibres and ensure competitive wound exudate management capability. To investigate the effect of the fibrous configuration, this two-layer electrospun system was compared with a single-layer membrane of core-shell fibres made of PAA fibre core and PMMA-co-MAA fibre shell. Electrospun samples were characterised with regards to fibre characteristics (via electron microscopy), dye loading/release capability (via UV-Vis spectroscopy), colour change (via digital macrographs) and chemical composition (via attenuated total reflectance-Fourier transform infrared spectroscopy, ATR-FTIR).

## 2. Materials and Methods

### 2.1. Materials

PAA (*M_v_*: 450,000 g·mol^−1^), tetraethylene glycol (TEG), sulfuric acid (H_2_SO_4_), N,N-dimethyl-formamide (DMF, 99.98%), isopropyl alcohol (IPA, ≥99.7%), sodium hydrogen phosphate (Na_2_HPO_4_), citric acid and dimethyl sulfoxide (DMSO) were purchased from Sigma-Aldrich (Gillingham, UK). PMMA-co-MAA (*M_w_*: 125,000 g·mol^−1^; [MAA]:[MMA] = 1:2) was kindly supplied by Evonik Rohm GmbH (Weiterstadt, Germany). BTB was purchased from Alfa Aesar (Heysham, UK). All chemicals and solvents were used as received.

### 2.2. Preparation of Electrospinning Solutions

Polymer and BTB-loaded electrospinning solutions were prepared as follows. PMMA-co-MAA was dissolved (13% w/v) in a mixture of IPA and DMF (8:2 v/v), whilst a solution of PAA (with either 10 or 12 wt.% PAA) was prepared in a mixture of distilled water and IPA (8:2 v/v) with gently magnetic stirring (25 °C, 24 h). BTB-loaded solutions of PMMA-co-MAA were prepared by adding 2 wt.% of BTB (with respect to the weight of the polymer in the solution), while stirring magnetically at 100 r·min^−1^ (25 °C, 24 h). TEG (15 wt.%) was added to the PAA electrospinning solution (with respect to the polymer weight), while stirring magnetically at 100 r·min^−1^ (25 °C, 6 h), aiming to enable post-spinning covalent crosslinking via heat-induced esterification. Prior to electrospinning, 1 M sulfuric acid was added to the TEG-loaded PAA solution at a volume ratio of 50 μL·ml^−1^ to promote the esterification reaction, as previously reported [74,75].

### 2.3. Free Surface Electrospinning of Two-Layer Fibrous Membrane

The two-layer fibrous membrane was obtained via sequential free surface electrospinning of PAA and PMMA-co-MAA solutions, using a NS LAB 200 Nanospider electrospinner (Elmarco, Liberec, Czech Republic) with a cylindrical electrode. The PAA (12 wt.%) solution (supplemented with TEG) was electrospun using 70 kV and 23 cm of electrostatic voltage and electrode-collector working distance, respectively. The rotation speed of the electrode and the linear speed of the polypropylene (PP) fabric collector (3 ± 0.3 denier) were 2 r·min^−1^ and 30 cm·min^−1^, respectively. The PMMA-co-MAA solution (supplemented with BTB) was electrospun against previously-formed PAA membrane, thereby yielding the second layer in resulting nanofibrous membrane. Free surface electrospinning of the PMMA-co-MAA solution was carried out with 80 kV, 23 cm and 2 r·min^−1^ of voltage, electrode-collector working distance and electrode rotation speed, respectively. Additionally, the linear speed of the PMMA-co-MAA fabric collector during FSES of PMMA-co-MAA was selected at either 10, 20 or 30 cm·min^−1^, so that two-layer electrospun membranes with varied thickness, area density and porosity could be obtained. Two-layer membrane samples were coded as ‘PAA(PMMA-co-MAA)X’, where ‘X’ indicates the PMMA-co-MAA collector linear speed selected during PMMA-co-MAA electrospinning, i.e., either ‘10′, ‘20′ or ‘30′.

### 2.4. Coaxial Electrospinning of Single-Layer Membrane Made of Core-Shell Fibres

Single-layer membranes of core-shell fibres were electrospun using a coaxial (bicomponent) spinneret (Inovenso Co., Istanbul, Turkey). The BTB-loaded PAA solution (with a PAA concentration of either 10 or 12 wt.%) was employed as the fibre core-forming electrospinning solution, whilst the PMMA-co-MAA solution (13% w/v) was employed as the fibre shell-forming solution. Each polymer solution was loaded into a 10 mL capacity syringe (Fisher Co., Loughborough, UK) and set up on a positive displacement microprocessor syringe pump (Model 200 Series, KD Scientific Inc., Holliston, MA, USA). The syringes were connected to the coaxial spinneret (inner diameter: 0.44 mm; outer diameter: 1.6 mm) via Teflon tubing in order to fabricate core-shell nanofibres with different shell thickness. The coaxial spinneret was connected to one electrode of a high voltage direct-current power supply, capable of generating up to 60 kV (EH60P1.5, Glassman, Pangbourne, UK). A flat metallic sheet (20 × 5 × 2 mm^3^) was covered with the polypropylene (PP) fabric (3 ± 0.3 denier) and grounded as the counter electrode to receive the electrospun core-shell nanofibres. Core-shell electrospinning was carried out with flow rates of 0.2 mL·h^−1^ (for the fibre core-forming solution) and 0.3 mL·h^−1^ (for the fibre shell-forming solution), electrostatic voltage of 25 kV, and a vertical needle-collector working distance of 15 cm. The temperature and relative humidity of the electrospinning environment in both electrospinning nanofibrous configurations were 25 °C and 40 r.h.%, respectively. The electrospun nanofibre membranes were dried at least two days at room temperature to remove any possible remained IPA, DMF or water prior to thermal treatment. Resulting membranes were coded as CS-PAAY-(PMMA-co-MAA), where ‘CS’ identifies the core-shell fibrous structure, whilst ‘Y’ indicates the polymer concentration of the fibre core-forming PAA solution, i.e., either ‘10′ or ‘12′.

### 2.5. Heat-Induced Network Formation in Two-Layer Electrospun Membranes

To induce the formation of the thermally-crosslinked PAA network, the two-layer electrospun membranes were incubated in a vacuum oven (130 °C, 85 kPa) for 30 min, and then cooled down to room temperature [74,75]. After this treatment, the resulting dry samples were weighted (*m_d_*) and incubated in water for 24 h in order to quantify the gel content of resulting thermally-crosslinked covalent network. Water-incubated samples were dried in vacuum at room temperature for 24 h and weighed (*m_e_*). The gel content was obtained according to Equation (1), as follows:(1)$$G=\frac{m_{e}}{m_{d}}\times 100$$

Thermally-crosslinked samples were coded as ‘PAA*(PMMA-co-MAA)X’, where ‘*’ identifies the crosslinked state of PAA fibres, whilst ‘X’ indicates the collector linear speed selected during electrospinning, as indicated in Section 2.3.

### 2.6. Membrane Density and Porosity Measurements

Two-layer electrospun membranes were characterised with regards to the area density, bulk density and porosity. The area density (*ρ_a_*) of electrospun nanofibrous membranes of known dimensions was measured via a semi-micro balance (Sartorius CP 225D, Goettingen, Germany), having a weighing accuracy of 10 nanograms. The bulk density (*ρ_b_*) was quantified by measuring the membrane thickness via a ProGage digital micrometer thickness tester (Thwing-Albert Instrument Company, West Berlin, NJ, USA) having a precision of 0.1 μm. The overall membrane porosity was indirectly calculated by assessing the hexadecane retention capacity (*C_PV_*) of the membrane, where hexadecane was employed as a low viscosity and surface tension liquid [76,77,78]. *C_PV_* was determined according to Equation (2):(2)$$C_{PV}=\frac{m_{sh}-m_{d}}{\rho \times m_{d}}$$
where *m_sh_* and *m_d_* are the masses of the hexadecane-saturated and respective dry membrane, respectively, whilst *ρ* is the density of hexadecane (*ρ* = 0.773 g·cm^−3^). Each measurement was replicated three times per sample and the mean and standard deviation were calculated and recorded.

### 2.7. Scanning Electron Microscopy (SEM)

SEM (Hitachi S-2600 N, Tokyo, Japan) was employed with a secondary electron detector to examine the surface and cross-section morphology of PMMA-co-MAA and PAA nanofibers obtained following electrospinning and thermal crosslinking, as well as respective two-layer membranes. All air dried samples were placed onto aluminium stubs (Ø: 12.7 mm) with the aid of carbon adhesive tapes and sputter gold-coated (Emitech K550X, London, UK) under high vacuum. An acceleration voltage of 3 kV was used with a typical working distance of 5.9–12.2 mm. Images were captured at different magnifications in the range of 100× to 10,000×. The average fibre diameters in each sample were estimated by evaluating a minimum of 50 fibres of the sample and were measured by means of image analysis software (SemAfore 5.21, JEOL CO., Helsinki, Finland) with at least 50 measurements per image to determine mean fibre diameter and associated frequency distributions. Values were expressed as mean ± standard error.

### 2.8. Transmission Electron Microscopy (TEM)

TEM (FEI Titan3 Themis 300, FEI^TM^, Hillsboro, OR, USA) was used to investigate the architecture core-shell electrospun fibres. Samples were collected onto a lacey carbon-coated copper grids with a 300 mesh. Images were taken at an acceleration voltage of 300 kV.

### 2.9. Attenuated Total Reflectance Fourier Transform Infrared (ATR-FTIR) Spectroscopy

ATR-FTIR spectra were acquired via a Spectrum BX spotlight spectrophotometer (Perkin Elmer, Llantrisant, UK) equipped with a diamond ATR attachment system. Scans were taken in the range of 4000–600 cm^−1^ and 64 repetitions were averaged for each spectrum sample. The resolution was 4 cm^−1^ and the scanning interval was 2 cm^−1^.

### 2.10. Swelling Measurements

Nanofibrous membranes of thermally-crosslinked PAA fibres were incubated in varied buffers (pH 5–8, 10 mL, 25 °C). For the determination of the water uptake (*WU*), samples were collected at selected time points, paper blotted and weighed. *WU* was calculated according to Equation (3):(3)$$WU=\frac{m_{t}-m_{d}}{m_{d}}\times 100$$
where *m_d_* is the mass of the freshly-prepared dry sample and *m_t_* is the mass of the corresponding hydrated sample.

Together with *WU*, the swelling index (*SI*) of thermally-crosslinked samples of known size (8 × 8 mm^2^) was measured by recording the change in sample surface area, as described in Equation (4):(4)$$SI=\frac{A_{t}}{A_{d}}\times 100$$
where *A_t_* is the surface area of the hydrated sample at the selected time point of incubation, whilst *A_d_* is the surface area of the dry, freshly-prepared sample.

### 2.11. Quantification of BTB Loading and Release Capability of Nanofibrous Membranes

McIlvaine buffer was chosen to simulate pH condition of healing and infected wounds (pH: 5–8) [79]. Prior to the quantification of the release capability, the amount of BTB encapsulated in the two-layer and core-shell electrospun membranes was quantified (prior to thermal treatment) by dissolving individual samples in 4 mL DMSO. An aliquot of the resulting solution (0.5 mL) of was added to 4.5 mL of the McIlvaine buffer solution. The solution was loaded in a glass cuvette with an optical length of 1 cm and the solution absorbance was recorded (Jasco V-630, JASCO GmbH, Easton, MD, USA) at either 433 or 616 nm depending on whether a buffer at pH 5–6 or 7–8 was employed. The amount of BTB encapsulated in the electrospun nanofibrous membranes was therefore calculated using a calibration curve built at the corresponding solution pH, whereby the presence of DMSO in the solution proved to minimally affect the UV-Vis readings.

Following quantification of BTB loading, 0.65–1.5 mg of BTB-loaded electrospun samples were immersed in 3 mL of the McIlvaine buffer solution at room temperature. At selected time points (0–96 h), 3 mL of the supernatant was loaded to the glass cuvette. The amount of BTB in each solution was determined via UV-Vis spectrophotometry at the same wavelengths as previously mentioned. Resulting absorbance data were used to derive the amount of BTB released from the samples at each time point and solution pH. The experiments were performed in triplicate, and average values were reported.

## 3. Results and Discussion

Figure 1 illustrates the overall research strategy pursued for the manufacture of infection-responsive membranes with varied fibrous configurations. PAA and PMMA-co-MAA were selected as membrane building blocks due to their well-known applicability in physiological environment, as well as due to their water absorbency and selective solubility in water, respectively. A two-layer membrane with integrated exudate management capability and infection responsivity was realised via sequential free surface electrospinning of TEG-loaded PAA and BTB-loaded PMMA-co-MAA solutions (Figure 1A).

Fibre loading with halochromic BTB dye in the PMMA-co-MAA layer was expected to equip the resulting membrane with colour change capability following contact with infected exudate (i.e., at pH > 7). Furthermore, the selection of PMMA-co-MAA as building block of the infection-responsive top layer was dictated by the selective solubility of this polymer at basic pH. In light of this polymer behaviour in water, additional membrane-induced visual effects, other than colour change, were expected following contact with infected wounds (pH > 7), e.g., release of BTB away from, and increased volumetric swelling of, the membrane. Together with the infection responsivity, encapsulation of TEG in the PAA wound contact layer was key to achieve a crosslinked covalent network in respective PAA fibres via esterification reaction of PAA carboxylic acid groups with TEG hydroxyl functions (Figure 1B). The synthesis of the covalent network in the wound contact layer was pursued to achieve membrane-induced exudate uptake following application situ.

Other than the two-layer fibrous configuration, a single-layer membrane of core–shell fibres was also investigated via coaxial electrospinning of fibre core-forming BTB-loaded PAA solution and fibre shell-forming PMMA-co-MAA solution (Figure 1C). In comparison to the former membrane, the single-layer membrane of core-shell fibres was designed to investigate the pH-sensitivity of the PMMA-co-MMA fibre shell, on the one hand, and the release capability of BTB-loaded PAA fibre core, on the other hand. In the following, the morphology and molecular organisation of electrospun membrane and respective fibres will be inspected via electron microscopy and ATR-FTIR spectroscopy, whilst membrane exudate management, dye release and colour change capabilities will be investigated in vitro at varied pH by combining swelling measurements with UV-Vis spectroscopy.

### 3.1. Fabrication of Two-Layer Nanofibrous Membrane

Free surface electrospinning of both PAA and PMMA-co-MAA solutions was successfully demonstrated with a cylindrical electrode partly immersed in the polymer solution (PS) (Figure 1A). In this setup, the electrode faced a ground-connected collector plate carrying a meltspun PP nonwoven fabric, which rotated perpendicularly to the fibre-forming electrospinning jet. Our previous work demonstrated that the cylindrical electrode proved to provide increased yield of fibre formation and stability with respect to the four wire electrode, at low polymer solution concentration [63,64]. At the selected voltage, electrode-collector distance and electrode rotation speed, two-layer electrospun membranes were prepared with minimal bead formation and homogeneous distribution of fibre diameters (Figure 2). 

In order to achieve controlled release of the BTB dye, variation in membrane porosity was pursued by controlling the linear speed of the fibre-collecting PAA layer during free surface electrospinning of the PMMA-co-MAA solution. Whilst in the case of PAA electrospinning the collector linear speed was kept constant at 30 cm·min^−1^, BTB-loaded PMMA-co-MAA fibres were collected at varying linear speed (*ν* = 10–30 cm·min^−1^), so that two-layer membranes with significantly different thickness (*h*: 36 ± 1 − 72 ± 2 μm), area density (*ρ_a_*: 1.27 ± 0.06 − 8.86 ± 0.05 g·m^−2^) and bulk density (*ρ_b_*: 35 ± 1 − 122 ± 3 kg·m^−3^) could be achieved (Table 2).

Control in the linear collector speed enabled us to vary the membrane bulk density and the porosity, which was important given their impact on diffusion and fluid transport phenomena in the fibrous structure. The porosity of resulting membranes was indirectly calculated by measuring respective capacity to retain hexadecane (*C_PV_*), so that increased *C_PV_* values were obtained with membranes electrospun at decreased collector linear speed (Table 2). Free surface electrospinning with varied collector linear speed is expected to generate fibrous layers with different porosity channels, which are critical to mediate the diffusion of BTB and aqueous medium throughout the structure. By decreasing the collector linear speed, there will inherently be increased time for the fibres to build thicker layers, so that membranes with decreased porosity and increased fibre density are realised. Together with the systematic variation in structural parameters, the presented design strategies could therefore enable the scalable formation of two-layer electrospun samples with controlled release capability.

The optical images of resulting electrospun samples seem to support previously-reported structural differences (Figure 1A), whereby membranes with increased density of BTB-loaded PMMA-co-MAA fibres displayed a yellow-like colour, in agreement with the BTB loading trends in the system. Other than the physical appearance, Figure 2 shows the SEM images of the cross-section morphologies observed in samples PAA*-PMMA-MAA, whereby the two-layer structure can be clearly detected. Sequential free surface electrospinning of PAA and PMMA-co-MAA solutions did not seem to cause any remarkable compression on the bottom layer, whilst membranes collected at decreased linear speed appeared somewhat denser, in line with previous measurements (Table 2). Other than the cross-section, the surface morphology of the electrospun fibres is shown in Figure 3. The average diameter of electrospun non-crosslinked PAA fibres was approximately 865 ± 315 nm with uniform fibre diameter distribution (Figure 3A), while thermal treatment proved to induce a decrease in fibre diameter (*d* = 680 ± 270 nm) (Figure 3B), as expected due to the material dehydration and the formation of a covalently crosslinked PAA network [80]. Other than the fibre diameter, PAA fibre surface morphology exhibited no significant difference after thermal treatment. In comparison, electrospun PMMA-co-MAA fibres also displayed a smooth surface with no visible bead, although slightly larger diameters (*d* = 1450 ± 610 nm) were measured (Figure 3C).

### 3.2. Synthesis of Thermally-Crosslinked Covalent Network in PAA Wound Contact Layer

Since PAA is a water soluble polymer, the synthesis of a covalently-crosslinked PAA network was pursued to ensure water-insolubility and retention of fibrous structure in the electrospun PAA layer. To minimise the risk of solvent-induced alteration of fibre architecture during crosslinking reaction, either TEG-loaded PAA electrospun fibres or two-layer electrospun membranes were incubated in vacuum (85 kPa) at 130 °C for 30 min [74,75], to induce TEG-initiated esterification reaction of PAA carboxyl groups (Figure 1C). After thermal incubation, resulting PAA nanofibres appeared tougher with respect to the pristine material and displayed some shrinking. To confirm the presence of a covalent network, thermally-incubated samples were immersed in distilled water so that the gel content was quantified. No significant gravimetric changes were recorded in the retrieved dry samples, so that *G* values of at least 98 wt.% were measured. These results confirmed the formation of a covalently-crosslinked PAA network in the electrospun fibres, so that respective wound contact layer proved to be water-insoluble in aqueous environment (regardless of the solution pH) and to display retained fibre structure following hydration, in contrast to the case of non-thermally-treated samples.

Together with the evaluation of the gel content, thermally-treated PAA fibres were also analysed with regards to their swelling behaviour. Swelling measurements were carried out via gravimetric and surface area analyses in varied buffer solutions (pH = 5–8), to further elucidate the molecular organisation of resulting fibres, on the one hand, and to quantify the aqueous medium management capability of the ultimate two-layer membrane, on the other hand. Following incubation in water, samples PAA* turned out into a gel-like and nearly translucent material, whilst exhibiting a pH-sensitive water uptake (*WU*) and swelling index (*SI*) (Figure 4). Whilst PAA fibres recorded an almost 13-fold averaged increase in weight at pH 5 (*WU* = 1291 ± 48 wt.%), an almost doubled averaged *WU* was measured at pH 8 (*WU* = 2369 ± 34 wt.%), whilst intermediate values were observed within this pH range (Figure 4A). The high compatibility of PAA fibres with aqueous environment was confirmed by the fact that sample equilibration with water was reached within 4 and 8 min at pH 5 and 8, respectively. Consequent to the uptake of aqueous medium, samples also displayed an instantaneous, significant increase in surface area, so that an *SI* of 272 ± 4 and 285 ± 3 a.% was measured at pH 5 and 8, respectively (Figure 4B). The gradual increase of *WU* and *SI* described by respective temporal profiles is in line with the presence of a covalently-crosslinked polymer network, which mediates the diffusion of buffer molecules and salt ions into the system [78,80,81]. The fact that fibres displayed increased values of *WU* and *SI* following incubation at increased pH agrees with the presence of negatively-charged carboxylic acid groups in fibre-forming PAA chains (*pK_a_* = 4.5) [82], suggesting that there are still free carboxylic acid groups in the system following network formation. As the solution pH is increased from 5 to 8, the repulsion between the negatively-charged carboxylic groups of PAA chains leads to increased free volume for water molecules to diffuse into the fibres [78,80,83,84]. These results therefore support the use of thermally-crosslinked PAA fibres as wound contact layer, in light of their ability to absorb water, which is key to ensure a moist environment in situ [67].

### 3.3. Morphology of Single-Layer Membranes of Core-Shell Electrospun Fibres

Other than the design of two-layer membrane, single-layer core-shell fibres were also investigated as an alternative infection-responsive fibrous configuration. As described in Figure 1B, a coaxial spinneret was used to obtain fibres made of a BTB-loaded PAA core and a PMMA-co-MAA shell, with varied shell thickness. Due to inherent challenges associated with the formation of homogeneous core-shell electrospun fibres [85], electrospinning parameters were carefully adjusted in order to achieve continuous coating of the polymer-forming shell onto the fibre core: (i) the electric field strength of approximately 1.6 kV·cm^−1^ enabled minimised interfacial instability and the formation of a single Taylor cone including both the core- and the shell-forming solutions, rather than multiple jets and spinning of separate, non-coaxial fibres [86]. (ii) The selected flow rate of the core solution was lower than the shell solution in order to avoid intermingling of the core and shell solutions and facilitate the formation of an inner jet and bead-free uniform fibres [85,86,87]. (iii) The polymer concentration (10–12 wt.% PAA; 13 wt.% PMMA-co-MAA) and the viscosity of the core and shell solutions were also adjusted to avoid the intermingling between core and shell solutions [87].

Despite the fact that homogeneous core-shell fibres were successfully obtained, the shell wall thickness of the resulting electrospun fibres did not display any detectable variation when the flow rate of the core solution was varied during coaxial electrospinning, as suggested elsewhere in the literature [86,87,88,89]. Due to the complex nature of the coaxial electrospinning, we hypothesise that varying the flow rate of the core solution did have detrimental impact on the co-spinnability of the selected solutions [87], thereby negatively impacting on the drug delivery and infection-triggered colour change capabilities of resulting fibres [90]. In order to control the shell wall thickness, we varied the PAA concentration in the BTB-loaded core solution, while maintaining the polymer concentration in the shell solution and the geometry of the coaxial spinneret constant. Based on the SEM and TEM images (Figure 5), the averaged overall diameter of the core-shell fibres did not vary by increasing the PAA concentration in the core solution, in line with previous reports [90], partially due to the little variation in selected polymer concentrations (~10–12 wt.%). On the other hand, the diameter of the fibre core (and inversely the shell wall thickness) proved to increase in fibres electrospun with increased PAA concentration in the fibre core-forming solution, as indicated in Figure 5B,C. Samples CS-PAA10-(PMMA-co-MAA) displayed the smallest fibre core diameter (*d_c_* = 0.43 ± 0.07 µm), i.e., thicker wall, whilst this was increased to 0.555 ± 0.035 µm when the PAA concentration was increased from 10 to 12 wt.% in the fibre core-forming solution, with no significant variation in the overall fibre diameter (*d* = 2 ± 0.5 µm).

The above-described direct relation between polymer concentration in the core-forming solution and fibre core diameter can be imputed to the evaporation of solvent during the flying time in the electrospinning process and the quick drying of the shell solution comparatively to the fibre core-forming solution [87,91].

### 3.4. Elucidation of the Chemical Composition via ATR-FTIR Spectroscopy

ATR-FTIR spectroscopy was employed in order to elucidate the chemical composition of both two-layer membranes and single-layer core-shell fibres. Figure 6 presents the ATR-FTIR spectra of pure BTB raw material, electrospun PAA, thermally-crosslinked electrospun PAA, electrospun PMMA-co-MAA and core-shell fibres CS-PAA12-(PMMA-co-MAA). The FTIR spectrum of raw BTB exhibits bands at 1346 cm^−1^, 1162 cm^−1^ and 1042 cm^−1^, which are assigned to the –SO_3_ functional group and the asymmetric and symmetric S-O-C stretching vibration bands at 880 cm^−1^ and 796 cm^−1^ [92,93]. A band at 651 cm^−1^ due to C–Br stretching vibrations is also detected [94]. The FTIR spectra of electrospun PAA nanofibres were characterised by the stretching vibrations of carboxylic acid functional groups at 1700 cm^−1^, a broad band at 2700-3300 cm^−1^ related to hydroxyl and carboxylic functional groups and overlapping with the –CH_2_ groups at 2946 cm^−1^ [95]. The peak at 1172 cm^−1^ is attributed to C–O stretching mode in PAA, while the peak at 812 cm^−1^ is due to the O–H out of plane motion of the carboxylic groups in PAA [96]. Other than the electrospun PAA sample, the FTIR spectra of the thermally-crosslinked variant appeared roughly similar. The –C=O stretching vibration at 1695 cm^−1^ was increased after heat treatment and shifted to lower frequencies, likely due to the formation an ester bond between TEG hydroxyl and PAA carboxylic acid groups [97]. However, no clear band associated with ester linkages was detected, in agreement with the published literature [98], which is attributed to the overlap with the carboxylic acid-related band. Other than PAA samples, the FTIR spectrum of the electrospun PMMA-co-MAA fibres exhibits characteristic bands of methyl and methylene C–H stretching vibrations at 2990 cm^−1^, a strong and intensive band of carboxylic acid groups at 1727 cm^−1^, and two bands at 1242 and 1148 cm^−1^ depicting the presence of ester linkages (C–O–C stretching) [99]. Ultimately, the spectrum of sample CS-PAA12-(PMMA-co-MAA) indicates that the BTB is well dissolved and stabilised in the host matrix without deterioration of active groups, indicating good compatibility between the dye and the fibre components.

### 3.5. Loading and Release Capability of Electrospun Nanofibrous Membranes

Electrospinning of BTB-loaded PMMA-co-MAA solutions enabled the formation of dye-encapsulated fibres with 86 wt.% of encapsulation efficiency. BTB release assessments were initially performed with the two-layer electrospun membranes of varied porosity and consisting of the thermally-crosslinked PAA wound contact layer (*ρ_a_* = 0.78 ± 0.5 g·m^−2^) and BTB-loaded PMMA-co-MAA infection-responsive top layer (*ρ_a_* = 36.3 ± 0.8 − 72.4 ± 1.9 g·m^−2^). Despite all samples proved to display full release of BTB in alkaline conditions, variation in top layer porosity proved to impact on the release profile of the dye (Figure 7). In the case of sample PAA*-PMMA-co-MAA30 (Figure 7A), almost 99 wt.% of encapsulated BTB dye was released within 2 h at both pH 7.4 and 8, whereas a slightly slower release was observed at pH 7, confirming the fact that not all the PMMA-co-MAA fibres were fully dissolved in these conditions. Overall, the temporal profiles show an initial burst release of BTB following incubation of all two-layer electrospun membranes. In acidic environment, the averaged amount of BTB released within 4 h was up to nearly 14 wt.%, whilst almost 74 wt.% BTB dye was still retained in the samples even following 96-h incubation. These results therefore confirm the barrier function of PMMA-co-MAA fibres at selected pH 5–6. When samples PAA*-PMMA-co-MAA10 with increased area density were analysed, the BTB dye release was somewhat decreased to 23 ± 1 wt.% in the same time interval, confirming the direct relationship of membrane porosity with soluble factor diffusion. 

Consequently, fibre collection at decreased linear speed during free surface electrospinning proved to generate membranes with decreased porosity, providing the BTB molecules with decreased free volume and diffusion capability away from the fibrous structure [66,99]. Other than the two-layer fibrous configurations, the BTB release profile of core-shell electrospun fibres CS-PAA10-(PMMA-co-MAA) (*d_c_* = 0.43 ± 0.07 µm) and CS-PAA12-(PMMA-co-MAA) (*d_c_* = 0.56 ± 0.04 µm) are shown in Figure 8. Overall, the BTB release appeared significantly higher and faster than the one recorded with the two-layer membranes, regardless of the specific solution pH. Whilst fibre incubation in alkaline conditions promptly triggered complete dye release, an averaged release of 40–56 wt.% was recorded following 6-h incubation at pH 5–7, with increased values measured at increased pH. Interestingly, fibres with increased core diameter proved to exhibit faster release with respect to fibres with decreased core diameter. Given that the overall fibre diameter was kept constant among the two samples (*d* = 2.0 ± 0.5 µm), this observation is attributed to the fact that the variation in fibre core diameter directly impacts on the dimension of the fibre shell wall thickness and consequent barrier function of the PMMA-co-MAA shell [89,99]. Other than that, the increased dye release profiles displayed by the core-shell electrospun fibres may be attributed to local intermingling of the fibre core- and shell-forming solutions, resulting in the localisation of BTB molecules directly on the surface, rather than on the core, of the electrospun fibres, as reported previously [99]. Furthermore, the inherently smaller shell wall thickness in the core-shell fibres with respect to the thickness of the PMMA-co-MAA fibres in the previously-reported two-layer membranes (Table 2) may also account for the significantly-different release profile among the two fibrous configurations.

### 3.6. Infection Responsivity of the Two-Layer Nanofibrous Membrane in Vitro

Encapsulation of the halochromic BTB dye in the electrospun fibres could be exploited to achieve fibre-based devices capable of wound infection detection and monitoring during wound management. The two-layer electrospun membrane proved to be more suitable in triggering prompt colour change at infection-related pH with respect to the single-layer membrane of core-shell fibres, partially attributed to the potential structural non-homogeneity of the latter system. Figure 9 shows the early in vitro wound infection detection capability of the two-layer membrane following incubation in buffer solutions. Other than the colour change observed in BTB solutions at varied pH (Figure 9A) [50], Figure 9B–D show the corresponding response of sample PAA*PMMA-co-MAA10. When the sample was immersed in pH 5–6, a light yellow colour was visible, whilst a colour shift towards light lime, light turquoise and cyan blue was visible following incubation in alkaline conditions. These results demonstrate the pH responsivity of the two-layer electrospun membrane developed in this study and its potential applicability for infection diagnostics and remote wound monitoring. This study therefore supports the application of BTB dye as well as PAA and PMMA-co-MAA polymers for the scalable manufacture of intelligent sensors enabling both exudate management capability and visual infection responsivity.

## 4. Conclusions

With the spread of antibiotic resistance, wound infection can lead to wound chronicity, resulting in delayed healing and risks of gangrene and amputation. With the aim to support wound exudate management and minimise the use of antibiotics, we have manufactured a two-layer electrospun system providing visual indication of infection and capable to take up wound exudate when applied in situ. The two-layer fibrous configuration was successfully built via FSES of clinically-approved building blocks, i.e., PMMA-co-MAA, PAA and BTB, whereby selective polymer solubility was exploited to trigger colour change in alkaline pH, whilst covalent network synthesis was leveraged to enable water uptake and exudate management. The two-layer fibrous configuration proved to enable minimal dye release in acidic, non-infection-related pH, and drastic colour change at pH > 7. The effect of fibre configuration and wall thickness was explored in the dry and hydrated environment, yielding reliable release capabilities in non-/infection-related pH ranges. Further research directions will involve the adjustment of microscopic fibre organisation aiming to further control dry release profiles.

## Figures and Tables

**Figure 1 sensors-19-00991-f001:**
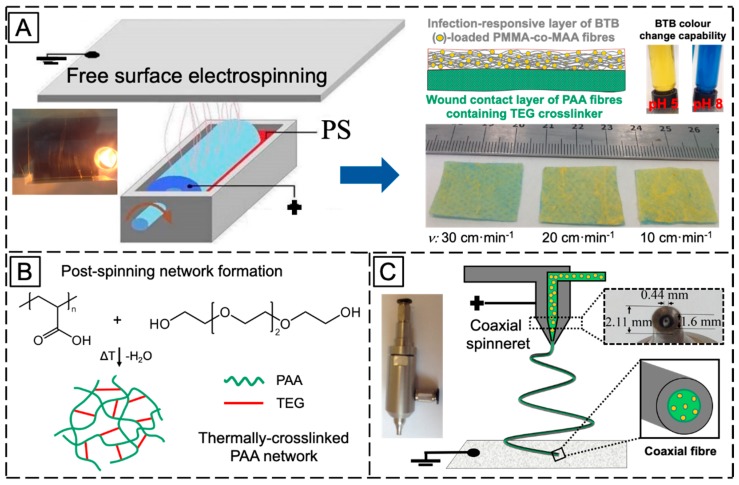
(**A**): Manufacture of the two-layer fibrous configuration via sequential free surface electrospinning of a PAA (

) wound contact layer containing the TEG crosslinker (

) and a BTB (
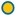
)-loaded PMMA-co-MAA (

) infection-responsive layer. BTB was introduced as halochromic dye to trigger colour change following infection-induced pH increase. BTB release was controlled by collecting PMMA-co-MAA fibres at varied collector linear speed (*ν*). (**B**): The two-layer system was thermally-treated to achieve a covalently-crosslinked network in PAA fibres via TEG-induced esterification. (**C**): Manufacture of the single-layer core-shell fibrous configuration via coaxial electrospinning of a fibre shell-forming PMMA-co-MAA solution and a BTB-loaded fibre core-forming PAA solution.

**Figure 2 sensors-19-00991-f002:**
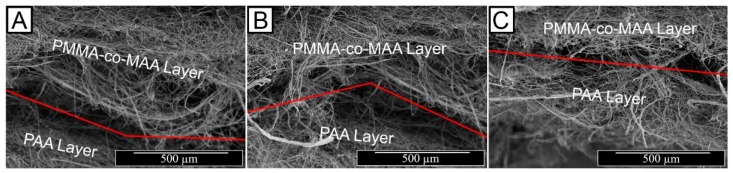
SEM images of the two-layer membrane cross-section displaying the thermally-crosslinked PAA wound contact layer and the BTB-loaded PMMA-co-MAA infection-responsive top layer. (**A**): PAA*(PMMA-co-MAA)30 (*ρ_a_*: 1.27 g·cm^−2^); (**B**): PAA*(PMMA-co-MAA)20 (*ρ_a_*: 3.88 g·cm^−2^; (**C**): PAA*(PMMA-co-MAA)10 (*ρ_a_*: 8.86 g·cm^−2^).

**Figure 3 sensors-19-00991-f003:**
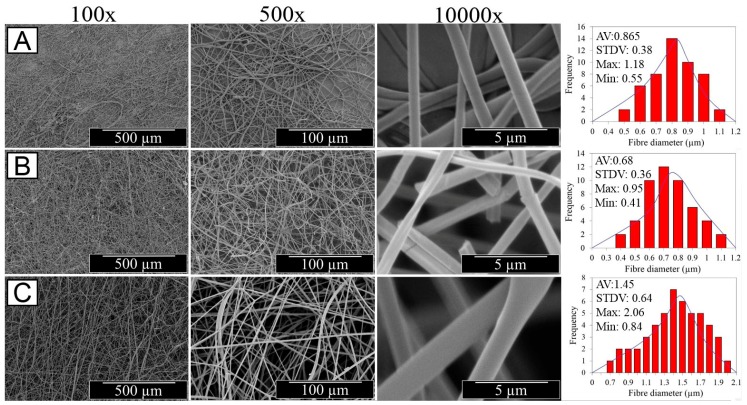
SEM images of PAA fibres following either free surface electrospinning (**A**) or thermally-crosslinked network formation (**B**), as well as of BTB-loaded PMMA-co-MAA fibres (*ν* = 10 cm·min^−1^) (**C**) at different magnifications. Respective averaged fibre diameter and standard deviation, as well as maximum and minimum fibre diameters are also provided (see Appendix A).

**Figure 4 sensors-19-00991-f004:**
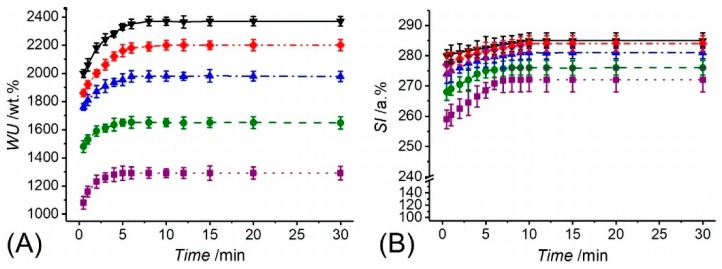
Water uptake (*WU*, **A**) and swelling index (*SI*, **B**) of thermally-crosslinked PAA fibrous membranes following incubation at varied pH. (··■··): pH 5; (--●--): pH 6; (─·▲·─): pH 7; (─··♦··─): pH 7.4; (─▼─): pH 8.

**Figure 5 sensors-19-00991-f005:**
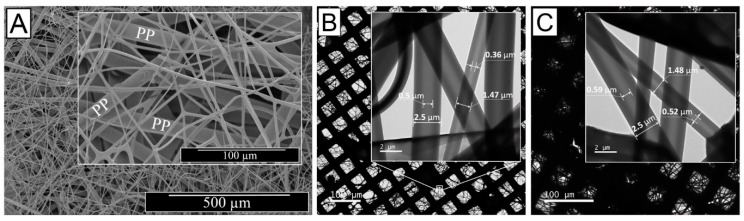
Microscopy images of single-layer membranes of core-shell electrospun fibres. (**A**) SEM image of CS-PAA12-(PMMA-co-MAA) fibres collected on a PP mesh. (**C**,**D**): TEM images of samples CS-PAA10-(PMMA-co-MAA) (**B**) and CS-PAA12-(PMMA-co-MAA) (**C**) once mounted onto a lacey carbon-coated copper grids. Inset images show individual fibres with core-shell boundaries.

**Figure 6 sensors-19-00991-f006:**
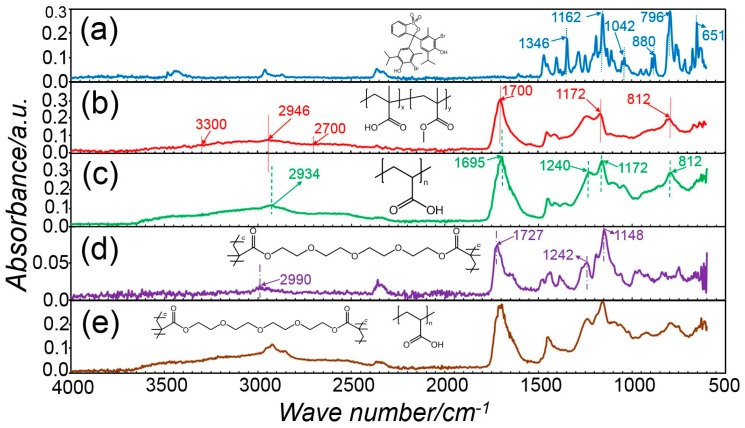
ATR-FTIR spectra of BTB raw material (**a**), PAA fibres (**b**), sample PAA* (**c**), PMMA-co-MAA fibres (**d**), and sample CS-PAA12-(PMMA-co-MAA) (**e**), with respective chemical formula.

**Figure 7 sensors-19-00991-f007:**
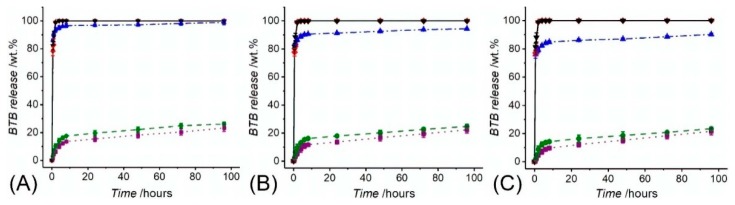
BTB release measured following incubation of samples PAA*(PMMA-co-MAA)30 (**A**), PAA*(PMMA-co-MAA)20 (**B**) and PAA*(PMMA-co-MAA)10 (**C**) at varied pH. (··■··): pH 5; (--●--): pH 6; (─·▲·─): pH 7; (─··♦··─): pH 7.4; (─▼─): pH 8.

**Figure 8 sensors-19-00991-f008:**
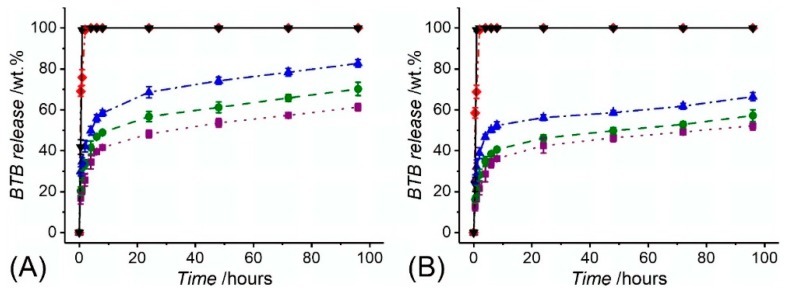
BTB release measured in varied buffer solutions from core-shell fibre samples CS-PAA12-(PMMA-co-MAA) (**A**) and CS-PAA10-(PMMA-co-MAA) (**B**). (··■··): pH 5; (--●--): pH 6; (─·▲·─): pH 7; (─··♦··─): pH 7.4; (─▼─): pH 8.

**Figure 9 sensors-19-00991-f009:**
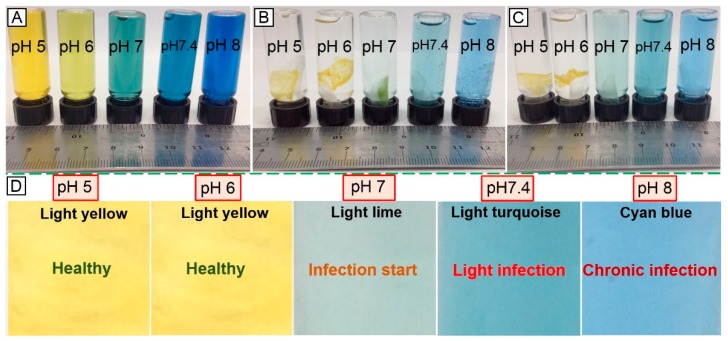
Visual infection responsivity of the two-layer BTB-loaded electrospun system. (**A**): Colouration of BTB solutions observed at varied pH. (**B**,**C**): Colour change capability of sample PAA*(PMMA-co-MAA)10 following contact with (**B**) and two-hour incubation in (**C**) aqueous medium at varied pH. At basic pH, PMMA-co-MAA fibres becomes soluble in aqueous medium so that fibre-loaded BTB is released, whilst inducing colour change. (**D**): Colour-infection maps obtained via membrane incubation at varied pH.

**Table 1 sensors-19-00991-t001:** State-of-the art design of wound pH sensors.

Design Strategy	Molecular Mechanism	Prototype Schematic	pH Range	Advantages	Limitations	Ref.
**- Grafting of pH indicator dyes to a substrate** **- Coating on fabrics or optical fibres**	- Dye absorbs specific light wavelengths depending on the environmental pH	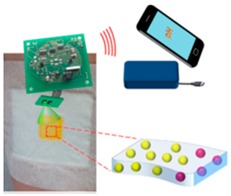	3.0–12.0	- Simple - Inexpensive	- Limited pH range sensitivity - Use of UV lamp required	[34,35,36,37,38,39,40]
**- Microfabrication of wire coils** **- Sandwich structure with a pH-sensitive hydrogel**	- pH-induced changes in hydrogel volume - Variation in proximal coil distance - Generation of inductance variation	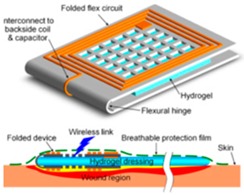	2.0–7.0	- Incorporation of several types of hydrogel possible via the sandwich structure	- Reading affected by changes in moisture content - Additional pH-insensitive hydrogel required - Most effective with a compression bandage	[41]
**- Sensitivity to wound exudate uric acid** **- Screen-printed surface electrode** **- Cellulose filter paper for** wicking and **fetching exudate** (to the detector)	- pH-sensitive oxidation capability of uric acid	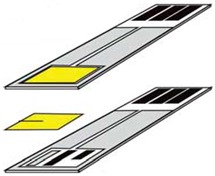	3.7–8.0	- Inexpensive - Disposable	- Not suitable for continuous wound monitoring - Decrease of signal quality over time due to electrode oxidation and biofouling	[42]
**- Immobilization of biological probes onto a polymer substrate**	- Variation of fluorescence emission intensity in response to pH changes	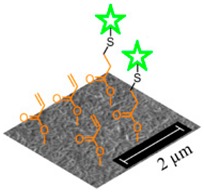	0.5–14.0	- Rapid response to pH	- Use of cytotoxic or physiologically-unstable materials - Camera-based devices necessary for fluorescence measurements	[43]
**- Dyeing of cotton swabs via incubation in the dyeing bath**	- Covalently dye immobilization to avoid wound contamination.	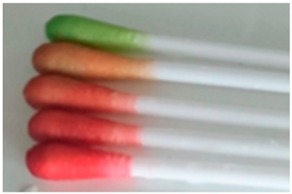	5.0–8.5	- Fast detection - Cost-effective	- Potential leaching of toxic dye species - Damage of the cell layer	[44]
**- Electrochemical sensor unit of gold-coated substrate or conductive polymer** **- Silver-based reference electrode**	- pH changes lead to absorption of hydrogen ions - Generation of an electrochemical potential with the reference electrode	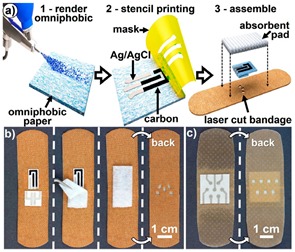	1.0–13.0	- Fast response over a wide pH range - High sensitivity - Long-term sensor reproducibility	- Direct contact of patients with electric current - Presence of potentially-toxic chemicals - Time-consuming and expensive design - Frequent calibration required	[7,26,48]

**Table 2 sensors-19-00991-t002:** Thickness (*h*), area (*ρ_a_*) and bulk (*ρ_b_*) density, as well as hexadecane retention capacity (*C_PV_*) of the two-layer nanofibrous membranes.

Sample ID	*h* [µm]	*ρ_a_* [g·m^−2^]	*ρ_b_* [kg·m^−3^]	*C_PV_* [µL·mg^−1^]
**PAA^* (a)^**	16 ± 1	0.78 ± 0.06	51 ± 2	14 ± 3
**PAA*(PMMA-co-MAA)30**	36 ± 1	1.27 ± 0.06	35 ± 1	21 ± 1
**PAA*(PMMA-co-MAA)20**	52 ± 1	3.88 ± 0.08	74 ± 1	20 ± 1
**PAA*(PMMA-co-MAA)10**	72 ± 2	8.86 ± 0.05	122 ± 3	18 ± 1

^(a)^ Single-layer membrane control of thermally-crosslinked PAA fibres.

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
