# Peer review of "Two-layer Electrospun System Enabling Wound Exudate Management and Visual Infection Response"

_sensors, 2019, doi:10.3390/s19050991_

Round 1
Reviewer 1 Report
In the introduction, we have the impression that the system presented will save the healthcare, I want to say that it is not so simple, the system even when technically performant will have to show medical evidence of its efficieny to be able to be accepted by the physicians and reimbursed by the health insurance. Secondly, the wound dressing is in contact with the open body and must not be dangerous for the patient, the chemicals must not contaminate the patient. It will be good to say a few word about the non-toxicity of the used chemicals.
The pH is an important parameter for the wound but I am not sure that alone it can be the universal indicator of the infection of the wound, the monitoring of several parameters is necessary to help the carers to give their diagnostics.
It could be good also to study the change of pH of the exudate when it move through the wound dressing to be sure that the indication given by the dressing is relevant for the value in the wound.
In the conclusion, the medical problem is simplified, the wound chronocity is a complex problem not only linked to antibiotic resistance but also to illness like diabete (diabetic foot ulcer) or blood circulation problem (venous leg ulcer) or pressure ulcer.
Author Response
Dear Editor,
Thank you very much for considering our manuscript entitled "An infection-responsive electrospun nanofibrous membrane with integrated colour change capability", and for the very useful feedback kindly provided by the referees. We have addressed each comment raised by the two reviewers (as reported below) and revised the manuscript accordingly with track changes. In the following referees’ questions have been addressed point-by-point, and, where applicable, changes reflected in the manuscript.
Point-by-point response to the referees’ comments
We would firstly like to thank the referees for their time in reviewing our manuscript; we believe the resubmitted version, amended in response to their comments, has strengthened the quality of our manuscript.
Reviewer 1
1) In the introduction, we have the impression that the system presented will save the healthcare, I want to say that it is not so simple, the system even when technically performant will have to show medical evidence of its efficiency to be able to be accepted by the physicians and reimbursed by the health insurance.
à Our answer: we thank the referee for raising this point, we feel that this question is mainly related to the potential clinical translation of the proposed device rather than on the respective design, manufacture and structure-function relationships. We agree that demonstrating clinical efficiency is key to ensure successful translation to human use. Rather than the clinical translation, however, our study has looked at and successfully demonstrated the scalable manufacture of a fibrous system with integrated wound exudate capability and visual infection responsivity via free-surface electrospinning using clinically-approved building blocks.
2) Secondly, the wound dressing is in contact with the open body and must not be dangerous for the patient, the chemicals must not contaminate the patient. It will be good to say a few word about the non-toxicity of the used chemicals.
à Our answer: we agree with the referee that the device toxicity is an important aspect that should be minimised aiming at translation to human use. In response to this point, we would like to emphasise that polyacrylic acid (PAA), poly(methyl methacrylate-co-methacrylic acid) (PMMA-co-MAA) and bromothymol blue (BTB) were employed in the study as clinically-approved, non-toxic building blocks. BTB was encapsulated in the fibres to equip resulting prototype with colour change capability.
To reflect this point, a sentence about the clinical applicability of these materials has been added at the end of the introduction (L139 – L142), and reads:
“The use of PAA, PMMA-co-MAA and BTB was also supported by the fact that these materials are considered non-hazardous substances according to European regulation EC 1272/2008, so that the infection-induced BTB release would not negatively impact on wound healing”.
3) The pH is an important parameter for the wound but I am not sure that alone it can be the universal indicator of the infection of the wound, the monitoring of several parameters is necessary to help the carers to give their diagnostics.
à Our answer: we have discussed in the introduction (L71 – L73) that different wound biomarkers have been reported for wound monitoring purposes. Among these, there is robust published literature supporting the measurement of wound pH to diagnose the occurrence of infection [7, 11]. Based on this and on the simplicity of these measurements, we focused our attention on the pH as wound infection biomarker, aiming to design an integrated fibrous system with integrated exudate management capability and visual infection responsivity.
4) It could be good also to study the change of pH of the exudate when it moves through the wound dressing to be sure that the indication given by the dressing is relevant for the value in the wound.
à Our answer: we agree with referee that testing with wound exudates is a key step to investigate the prototype functionalities in nearly-physiologic conditions, prior to moving to human trials. We are in the process of starting this work and will report on this in due course.
5) In the conclusion, the medical problem is simplified, the wound chronicity is a complex problem not only linked to antibiotic resistance but also to illness like diabetic (diabetic foot ulcer) or blood circulation problem (venous leg ulcer) or pressure ulcer.
à Our answer: we take referee’s point on the complexity to control wound chronicity, although the biomarkers selected in this study have been supported by a wide range of publications. In line with the referees, we have revised the conclusions, as reported below:
“With the spread of antibiotic resistance, wound infection can lead to wound chronicity, resulting in delayed healing and risks of gangrene and amputation. With the aim to support wound exudate management and minimise the use of antibiotics, we have manufactured a system providing visual indication of infection and capable to take up wound exudates when applied in situ. Two fibrous configurations were successfully built via FSES using clinically-approved building blocks, i.e. PMMA-co-MAA, PAA and BTB, whereby selective polymer solubility was exploited to trigger colour change in alkaline pH, whilst covalent network synthesis was leveraged to enable water uptake and exudate management. The two-layer fibrous configuration proved to enable minimal dye release in acidic, non-infection-related pH, and drastic colour change at pH >7. The effect of fibre configuration and wall thickness was explored in the dry and hydrated environment, yielding reliable release capabilities in non-/infection-related pH ranges. Further research directions will involve the adjustment of microscopic fibre organisation aiming to further control dry release profiles.”
In line with the scope of this study, the title of the manuscript has been revised and now reads:
“Manipulation of electrospun fibre configuration in a wound dressing system with integrated wound exudate capability and visual infection responsivity”.
Reviewer 2 Report
This manuscript describes an electrospun fibrous membrane with pH responsiveness and liquid absorption capabilities. The membrane may be potentially used for wound management. However, some parts of the manuscript are overstated or inaccurately described. For example, the title of the manuscript says “nanofibrous membrane”. However, according to the microscopic images, the diameter for most of the fibers is larger than 1 micron, so it is more accurate to say “microfibrous membrane”. On line 489-492, it says “These results demonstrate that the naofibrous membrane can be used as a diagnostic marker for wound status … to the healing process.” As no clinical trial has been conducted on patient subjects to show that the membrane indeed behaves as expected, it is still too early to be so conclusive.
In pH responsive membrane, the fluorophore is usually covalently immobilized in the membrane so that the dye won’t leach out while the sensor is being used. In this manuscript, the membrane will dissolve and release the dye upon contact with a basic solution. Why does the dye need to be released? Are there any advantages for this? Do the dissolution of the membrane and the release of the dye have any potential effect on the wound?
There are some typos or minor errors in the manuscript. For example, on line 491 “non-invasively detector” should be “non-invasive detector”. Please edit the manuscript carefully before resubmitting it.
Author Response
Dear Editor,
Thank you very much for considering our manuscript entitled "An infection-responsive electrospun nanofibrous membrane with integrated colour change capability", and for the very useful feedback kindly provided by the referees. We have addressed each comment raised by the two reviewers (as reported below) and revised the manuscript accordingly with track changes. In the following referees’ questions have been addressed point-by-point, and, where applicable, changes reflected in the manuscript.
Point-by-point response to the referees’ comments
We would firstly like to thank the referees for their time in reviewing our manuscript; we believe the resubmitted version, amended in response to their comments, has strengthened the quality of our manuscript.
Reviewer 2
1) This manuscript describes an electrospun fibrous membrane with pH responsiveness and liquid absorption capabilities. The membrane may be potentially used for wound management. However, some parts of the manuscript are overstated or inaccurately described. For example, the title of the manuscript says “nanofibrous membrane”. However, according to the microscopic images, the diameter for most of the fibers is larger than 1 micron, so it is more accurate to say “microfibrous membrane”.
à Our answer: we agree with the referee about the points raised on the title with respect to the fibre diameters measured in our prototypes. To include this comment and better capture the scope of the study, the title has been revised and now reads:
“Manipulation of electrospun fibre configuration in a wound dressing system with integrated wound exudate capability and visual infection responsivity”.
2) On line 489-492, it says “These results demonstrate that the naofibrous membrane can be used as a diagnostic marker for wound status … to the healing process.” As no clinical trial has been conducted on patient subjects to show that the membrane indeed behaves as expected, it is still too early to be so conclusive.
à Our answer: we agree with the referee that clinical trials are necessary to confirm the applicability of this system for human use. In line with the referee, that sentence has been deleted and the following sentence has been included:
“These results demonstrate the pH responsivity of the nanofibrous membrane developed in this study and its potential applicability for infection diagnostics and remote wound monitoring.”
3) In pH responsive membrane, the fluorophore is usually covalently immobilized in the membrane so that the dye won’t leach out while the sensor is being used. In this manuscript, the membrane will dissolve and release the dye upon contact with a basic solution. Why does the dye need to be released? Are there any advantages for this? Do the dissolution of the membrane and the release of the dye have any potential effect on the wound?
à Our answer: we take the point that the dye may be covalently-linked to the fibre-forming polymer to minimise risks of leaching, although this would add an additional synthetic step prior to fibre formation. Other than covalent coupling, dye leaching is minimised in this study by the manufacture of a bespoke fibrous architecture which prevents the dye release in healthy wound environment. Although infection triggers dye release from the fibres, the dye selected in this study is classified as a non-hazardous substance (EC 1272/2008) so that toxicity risks are minimised, promising well for the clinical applicability of the proposed prototype.
4) There are some typos or minor errors in the manuscript. For example, on line 491 “non-invasively detector” should be “non-invasive detector”. Please edit the manuscript carefully before resubmitting it.
à Our answer: we thank the referee for raising this point. The sentence above has been deleted and the manuscript has been proof-read and typos fixed.